The EMBO Journal (2013) 32, 2430–2438
www.embojournal.org

# A cyclic GMP-dependent signalling pathway regulates bacterial phytopathogenesis

**Shi-Qi An**[1,6]**, Ko-Hsin Chin**[2,6]**, Melanie Febrer**[3,6]**, Yvonne McCarthy**[4]**, Jauo-Guey Yang**[2]**, Chung-Liang Liu**[2]**, David Swarbreck**[5]**, Jane Rogers**[5]**, J Maxwell Dow**[4]**, Shan-Ho Chou**[2,*]** and Robert P Ryan**[1,*]

[1]Division of Molecular Microbiology, College of Life Sciences, University of Dundee, Dundee, UK, [2]Agricultural Biotechnology Center, Institute of Biochemistry, National Chung Hsing University, Taichung, Taiwan, ROC, [3]Division of Molecular Medicine, Colleges of Life Sciences, University of Dundee, Dundee, UK, [4]Department of Microbiology, Biosciences Institute, University College Cork, Cork, Ireland and [5]The Genome Analysis Centre, Norwich, UK

Cyclic guanosine 3′,5′-monophosphate (cyclic GMP) is a second messenger whose role in bacterial signalling is poorly understood. A genetic screen in the plant pathogen *Xanthomonas campestris* (*Xcc*) identified that *XC_0250*, which encodes a protein with a class III nucleotidyl cyclase domain, is required for cyclic GMP synthesis. Purified XC_0250 was active in cyclic GMP synthesis *in vitro*. The linked gene *XC_0249* encodes a protein with a cyclic mono-nucleotide-binding (cNMP) domain and a GGDEF diguanylate cyclase domain. The activity of XC_0249 in cyclic di-GMP synthesis was enhanced by addition of cyclic GMP. The isolated cNMP domain of XC_0249 bound cyclic GMP and a structure–function analysis, directed by determination of the crystal structure of the holo-complex, demonstrated the site of cyclic GMP binding that modulates cyclic di-GMP synthesis. Mutation of either *XC_0250* or *XC_0249* led to a reduced virulence to plants and reduced biofilm formation *in vitro*. These findings describe a regulatory pathway in which cyclic GMP regulates virulence and biofilm formation through interaction with a novel effector that directly links cyclic GMP and cyclic di-GMP signalling.

*The EMBO Journal* (2013) **32,** 2430–2438. doi:10.1038/emboj.2013.165; Published online 23 July 2013
*Subject Categories:* signal transduction; microbiology & pathogens
*Keywords:* biofilm; cyclic di-GMP; signal transduction; virulence; *Xanthomonas campestris*

## Introduction

Signal transduction pathways involving cyclic nucleotide second messengers occur in all domains of life where they act to link perception of environmental or intracellular cues and signal to specific alterations in cellular function. Bacteria use both cyclic mononucleotides and cyclic dinucleotides for this purpose (Linder, 2005; Pesavento and Hengge, 2009; Gomelsky, 2011). The roles of cyclic adenosine 3′,5′-monophosphate (cyclic AMP) and bis-(3′–5′)-cyclic di-guanosine monophosphate (cyclic di-GMP) in control of a variety of bacterial processes including virulence are now well established. By contrast, the roles in bacterial signalling of cyclic guanosine 3′,5′-monophosphate (cyclic GMP) and a second cyclic dinucleotide, cyclic di-AMP, are relatively poorly understood.

Although cyclic GMP has been detected in a number of organisms, compelling evidence for its role as a second messenger has only recently been obtained. Work on *Rhodospirillum centenum* has shown that cyclic GMP regulates the developmental process of cyst formation and has identified the guanylate cyclase involved in cyclic GMP synthesis (Gomelsky, 2011; Marden *et al*, 2011). Signal transduction leading to encystment involves a cyclic GMP-responsive transcription factor that is a homologue of CRP, the cyclic AMP-responsive transcription factor found in other bacteria. Beyond these observations, the distribution of cyclic GMP signalling in bacteria, the diversity of processes that are regulated and how regulation is exerted remain largely unknown. This lack of knowledge contrasts with the body of work on cyclic di-GMP signalling, which shows that in diverse bacteria this nucleotide regulates a range of functions including developmental transitions, biofilm formation, motility and virulence via interactions with different classes of effector molecule (Hengge, 2009; Sondermann *et al*, 2012; Srivastava and Waters, 2012; Ryan *et al*, 2012b; Romling *et al*, 2013).

Here, we have addressed the role of cyclic GMP signalling in *Xanthomonas campestris* pv. *campestris* (hereafter *Xcc*), the causal agent of black rot disease of cruciferous plants. As well as being a plant pathogen of global importance, *Xcc* is a model organism for molecular studies of plant–microbe interactions (Ryan *et al*, 2011; Mansfield *et al*, 2012). By analysis of transposon mutants of *Xcc*, we have identified a guanylate cyclase responsible for the synthesis of cyclic GMP both *in vivo* and *in vitro*. We show by mutational analysis that the cyclase is required for full virulence to plants and biofilm formation *in vitro*. We further show that these regulatory effects of cyclic GMP are exerted in part by a diguanylate cyclase (DGC) whose activity in cyclic di-GMP synthesis is responsive to cyclic GMP. This novel cyclic GMP effector thus represents a direct link between cyclic GMP and cyclic di-GMP signalling pathways.

## Results

### A genetic screen identifies that XC_0250 is required for cyclic GMP synthesis in Xanthomonas

In initial experiments, the synthesis of cyclic GMP by *Xcc* was detected by ELISA of a lysate of the wild-type strain grown in microtiter plates as described in Materials and methods. This

*Corresponding authors. S-H Chou, Agricultural Biotechnology Center, Institute of Biochemistry, National Chung Hsing University, Taichung, Taiwan, ROC. Tel.: +886 422840468; E-mail: shchou@nchu.edu.tw or RP Ryan, Division of Molecular Microbiology, College of Life Sciences, University of Dundee, Dundee DD1 4JU, UK. Tel.: +44 (0)1382384272; Fax: +44 (0)1382388216; E-mail: rpryan@dundee.ac.uk
[6]These authors contributed equally to this work.

protocol provided a basis for a screen for identification of components contributing to cyclic GMP synthesis. The wild-type *Xcc* strain 8004 was subjected to Mariner transposon mutagenesis, and mutants that gave reduced ELISA signals and contained a single and unique transposon insertion (determined by Southern analysis) were found from screening ∼5000 colonies. The region flanking the transposon insertion in each mutant was sequenced and compared with the sequenced genome of *Xcc* strain 8004. Two mutants were found to have the transposon inserted in different locations within *XC_0250*, which encodes a protein comprising a putative cyclase (CYC) domain linked to an uncharacterised domain with tetratricopeptide repeats (Figure 1; Supplementary Figure S1).

To confirm the contribution of *XC_0250* to cyclic GMP levels in *Xcc*, the level of the nucleotide was compared in wild type, a mutant in which the gene was deleted and a complemented mutant using HPLC (Supplementary Figure S1). In accordance with the previous findings, deletion of *XC_0250* led to a reduced cyclic GMP level, which was restored to wild type by complementation.

### The cyclase domain of XC_0250 is active in cyclic GMP synthesis

In order to demonstrate that XC_0250 is directly involved in cyclic GMP synthesis, the putative cyclase domain (amino-acid residues 1–220) was expressed as a recombinant protein with a C-terminal His6 tag and purified as described in Materials and methods. This protein had guanylate cyclase activity, converting GTP to cyclic GMP but had no activity as an adenylate cyclase on ATP (Figure 2 and data not shown). Comparison of the amino-acid sequence of the CYC domain of XC_0250 with characterised adenylyl cyclases (Shenoy and Visweswariah, 2004; Linder, 2005) identified the conservation of key residues. The two critical metal-ion binding aspartates are conserved (D41 and D71) as well as an alanine (A150) residue that occupies a substrate-specifying position. However, the transition state-stabilising asparagine and arginine residues are substituted by leucine (L157) and alanine (A161). The importance of both conserved and altered residues (D41, D71, L73, A150, L157 and A161) for the enzymatic activity of this domain was examined by assessing the effects of alanine or serine substitutions. Complementation of the *XC_0250* deletion mutant with clones expressing proteins with alanine substitutions in residues D41, D71, L73 and L157 failed to restore cyclic GMP levels to wild type (Supplementary Figure S1). Importantly, western analysis showed that all variant proteins were expressed similarly in *XC_0250* deletion mutant and wild-type backgrounds. Accordingly, the *in vitro* guanylate cyclase activity of these variants as well as A150S and A161S variants was also lost (Supplementary Figure S1). The findings indicated the critical nature of these residues for the enzymatic activity in cyclic GMP synthesis. Several of these residues (D71, L73, A150 and L157) are conserved in the *R. centenum* guanylyl cyclase (Supplementary Figure S1).

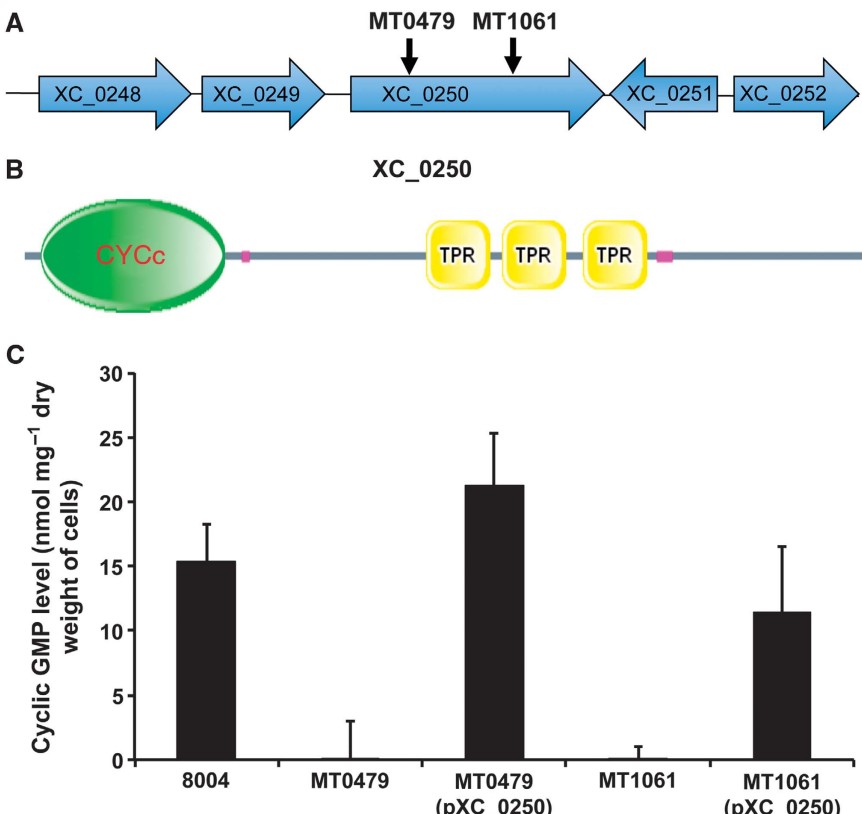

**Figure 1** Putative guanylyl cyclase in *Xcc* identified by transposon mutagenesis. (**A**) Genomic context of *XC_0250* and sites of Mariner transposon insertion (black arrows) that led to reduced cyclic GMP levels in *Xcc*. (**B**) Domain architecture of XC_0250 (886 aa) comprising an adenylyl/guanylyl cyclase catalytic domain (CYC) (aa 32–191) and three tetratricopeptide repeats (TPRs) (aa 413–480, 494–551 and 563–625). (**C**) Introduction of the *XC_0250* gene (cloned as pXC_0250) into the MT0478 and MT1061 Mariner transposon mutants restored cyclic GMP levels towards the level of the wild-type strain 8004. Values given are the mean and standard deviation of triplicate measurements (three biological and three technical replicates).

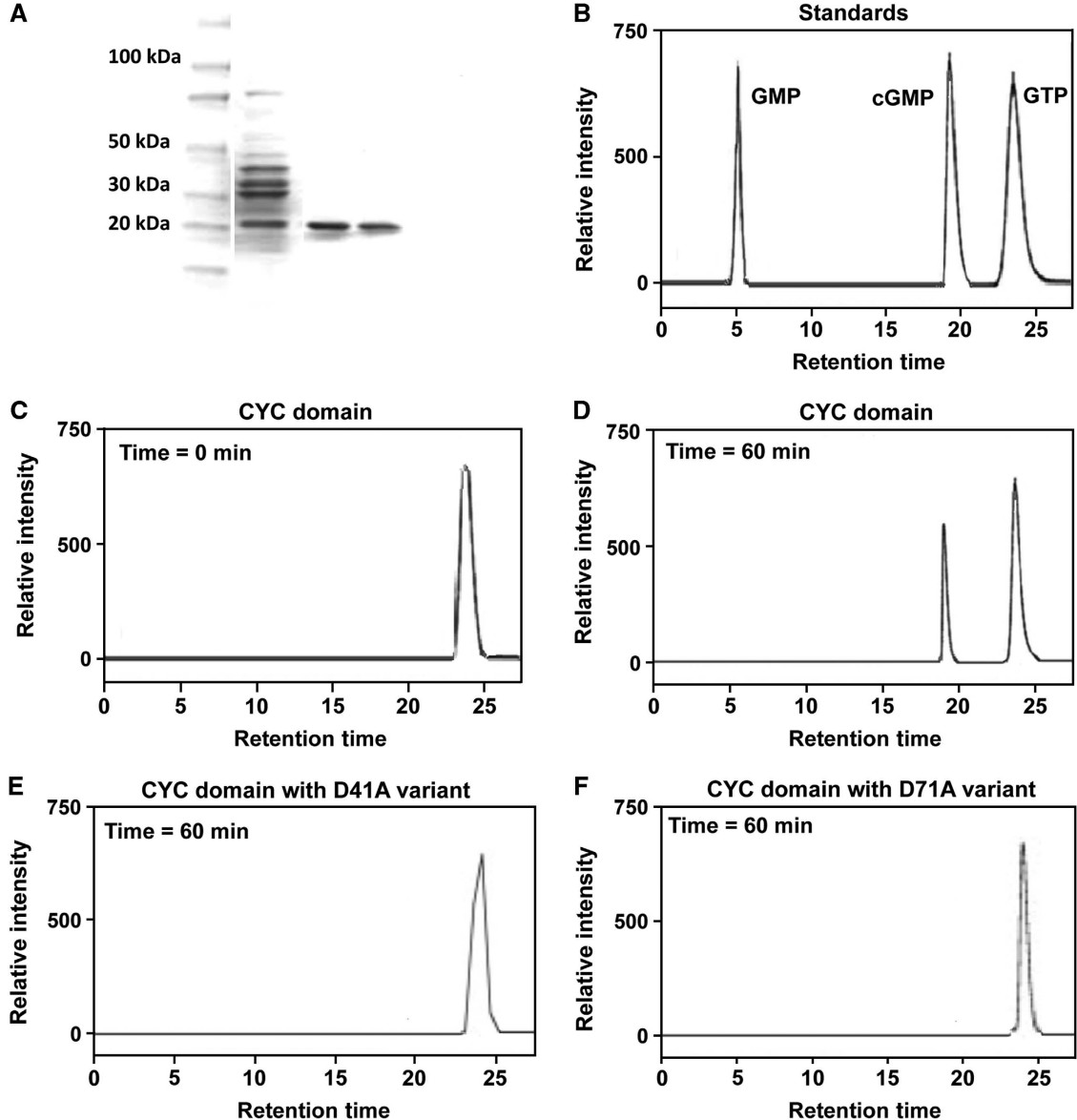

**Figure 2** The isolated CYC domain of XC_0250 possesses guanylyl cyclase activity. (**A**) SDS–PAGE of the CYCHis6 protein purified by nickel affinity chromatography showed a single band of the expected size of 21 kDa. Images shows spliced lanes from the SDS–PAGE gel—lane 1: protein marker; lane 2: total protein; lanes 3 and 4: samples purified by the Ni column method. The purified CYC domain from XC_0250 had enzymatic activity against GTP. (**B**) Reverse-phase HPLC separation of the reaction mixtures and guanine nucleotide standards GMP, cyclic GMP and GTP showed the synthesis of a compound with the same mobility as the cyclic GMP standard. (**C** and **D**) The CYC domain with mutations D41 or D71 variations lost enzymatic activity. (**E** and **F**) All enzymatic reactions contained 0.5 mM GTP, 10 mM MnCl$_2$ and 0.5 µg µl$^{-1}$ (0.025 nM) purified protein. Aliquots of reaction mixtures were boiled immediately after addition of the enzyme (time 0 min) and after 60 min of incubation (time 60 min). The identity of the product was confirmed by mass spectrometry.

### XC_0250 is required for maximal virulence to plants and biofilm formation

The role of cyclic GMP signalling in *Xcc* was investigated by comparative phenotypic and transcriptomic analyses of the wild-type and *XC_0250* deletion mutant. The mutant showed reduced virulence to Chinese Radish and reduced biofilm biomass when grown in complex media (Figure 3A and B). Complementation restored these phenotypes towards wild type (Figure 3A and B). Expression of the cyclase domain of XC_0250 alone could also restore biofilm formation to the *XC_0250* mutant, although expression of the enzymatically inactive D41A variant (see above) had no effect (Supplementary Figure S1).

Comparison of the transcriptome profiles of wild type and mutant by RNA-Seq showed that deletion of *XC_0250* led to alteration in levels of transcript of a number of genes (Figure 3C; Supplementary Table S1). These genes are associated with a range of biological functions that include bacterial motility and attachment, stress tolerance, virulence, regulation, transport, multidrug resistance, detoxification and signal transduction (Figure 3; Supplementary Table S1). The effect of *XC_0250* mutation on the level of transcript was validated by quantitative reverse transcription PCR (qRT–PCR) analysis for a panel of genes whose members were selected because of a previous implication in the virulence of *Xcc* (Figure 3C).

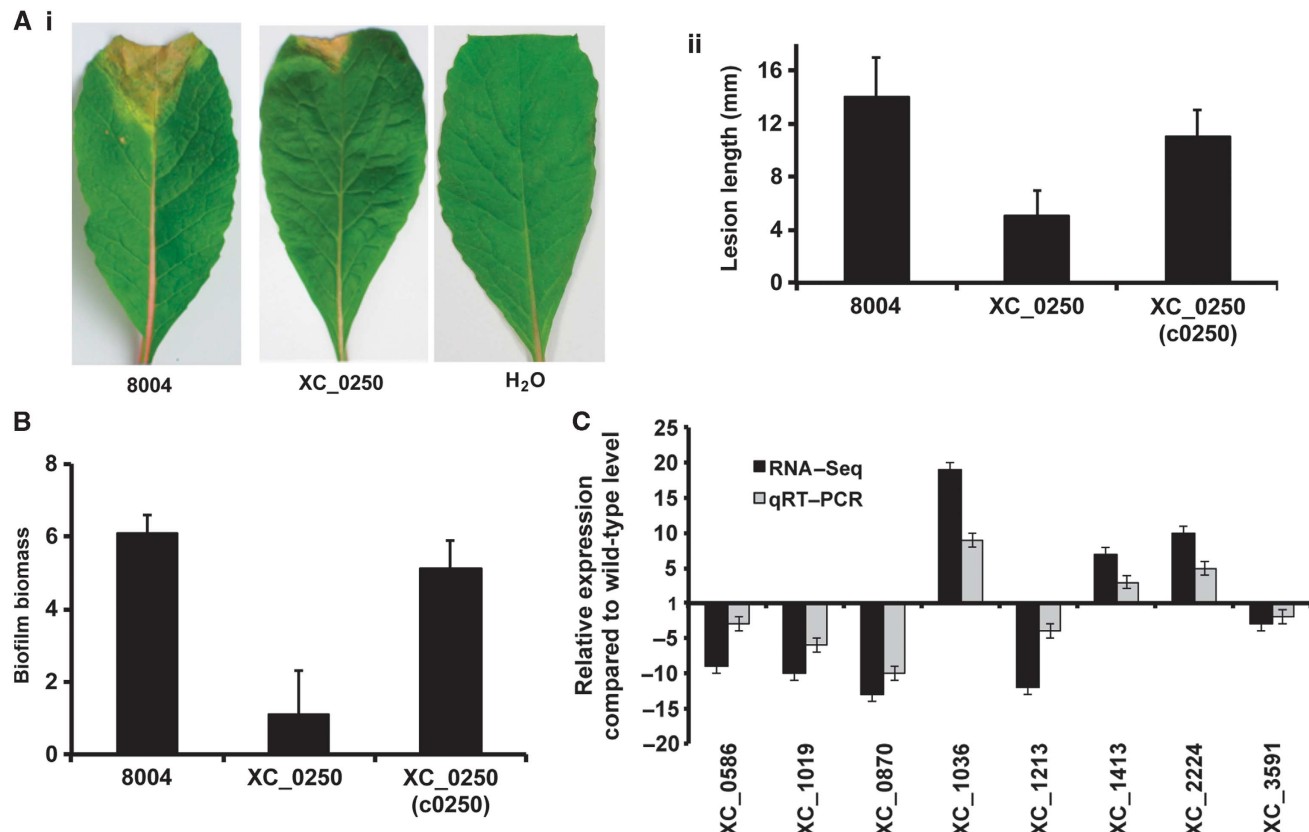

**Figure 3** Transcriptome and phenotypic characterisation of an *XC_0250* deletion mutant reveals roles in virulence and biofilm formation in *Xcc*. (**A**) The virulence of the *Xcc* strains was tested by measurement of the lesion length after bacteria were introduced into the vascular system of Chinese Radish by leaf clipping. (i) Representative virulence assays for (from left to right) *Xcc* wild type (8004), *XC_0250* deletion mutant and negative control (H₂O). (ii) Re-integration of the gene encoding *XC_0250* into a neutral site in the mutant chromosome (indicated as c0250) restores the reduced virulence of the mutant towards wild type. Values given are the mean and standard deviation of triplicate measurements (three biological and three technical replicates). (**B**) Deletion of *XC_0250* also led to decreased biofilm and cell adhesion on a glass surface when assessed by crystal violet staining (biofilm biomass is a measure of absorbance at 550/600 nm). Re-integration of the gene encoding *XC_0250* into the chromosome (indicated as c0250) restores the phenotypes towards wild type. Values given are the mean and standard deviation of triplicate measurements (three biological and three technical replicates). (**C**) Differential expression of selected genes implicated in virulence or biofilm formation in the *XC_0250* deletion mutant and wild type as determined by RNA-Seq (dark grey) and qRT–PCR (light grey). Mutation of *XC_0250* affected transcript levels of *XC_0586* (lipase), *XC_1019* (hypothetical protein), *XC_0870* (secreted protein), *XC_1036* (GGDEF domain), *XC_1213* (peptidase), *XC_1413* (chemotaxis protein), *XC_2224* (secreted protein) and *XC_3591* (pectate lyase). *In trans* expression of *XC_0250* (cloned as pXC_0250) in the mutant restored the gene expression towards wild type. The qRT–PCR data were normalised to 16S rRNA and is presented as the fold change with respect to the wild type for each gene. Data (means ± s.d.) are representative of four independent biological experiments. The complete RNA-Seq data set is detailed in Supplementary Table S1.

### XC_0249 is a cyclic GMP-binding protein

The effects of mutation of *XC_0250* on the range of phenotypes described above raise the issue of how cyclic GMP exerts a regulatory action in *Xcc*. One candidate cyclic GMP effector is XC_0249, a protein with a predicted cyclic mononucleotide-binding (cNMP) domain attached to a GGDEF domain that is active in synthesis of cyclic di-GMP (Supplementary Figure S2). The cNMP domain of XC_0249 was purified as described in Materials and methods. The ability of this domain to bind different cyclic mononucleotides and cyclic dinucleotides was assessed by isothermal titration calorimetry (ITC). The findings showed that the cNMP domain of XC_0249 bound cyclic GMP with a high affinity (a $K_D$ of $0.293 \pm 0.16\,\mu M$), but had a lower affinity for cyclic AMP ($17.9 \pm 2.02\,\mu M$), cyclic di-GMP or cyclic di-AMP (Figure 4A; Supplementary Table S2). Consistent with the above findings, the full-length XC_0249, the cNMP domain but not the GGDEF domain showed cyclic GMP binding when

assessed using the ³²P-labelled compound (Supplementary Figure S2).

To further characterise the binding of cyclic GMP by the cNMP domain of XC_0249, we determined the crystal structure of the domain with the nucleotide bound to a resolution of 2.2 Å. The XC_0249 cyclic NMP-binding domain crystallises as a dimer. The Fo-Fc electron density map of the complex (Figure 4) showed a clear electron density for cyclic GMP adopting a *syn* configuration when bound to the protein, that is with the ribose H1' and guanine H8 bonds oriented in the same direction. Cyclic GMP makes a number of contacts with the residues surrounding the ligand-binding pocket. A number of these residues form specific H bonds with the ribose 2'-OH, phosphate lactone oxygen atoms and, importantly, with the guanine base (Figure 4). Residue E146 is especially interesting, as it is located within the opposing αC helix, with its two carboxylate oxygen atoms forming two crucial H bonds with the guanine N1 and N2 atoms. N145 is

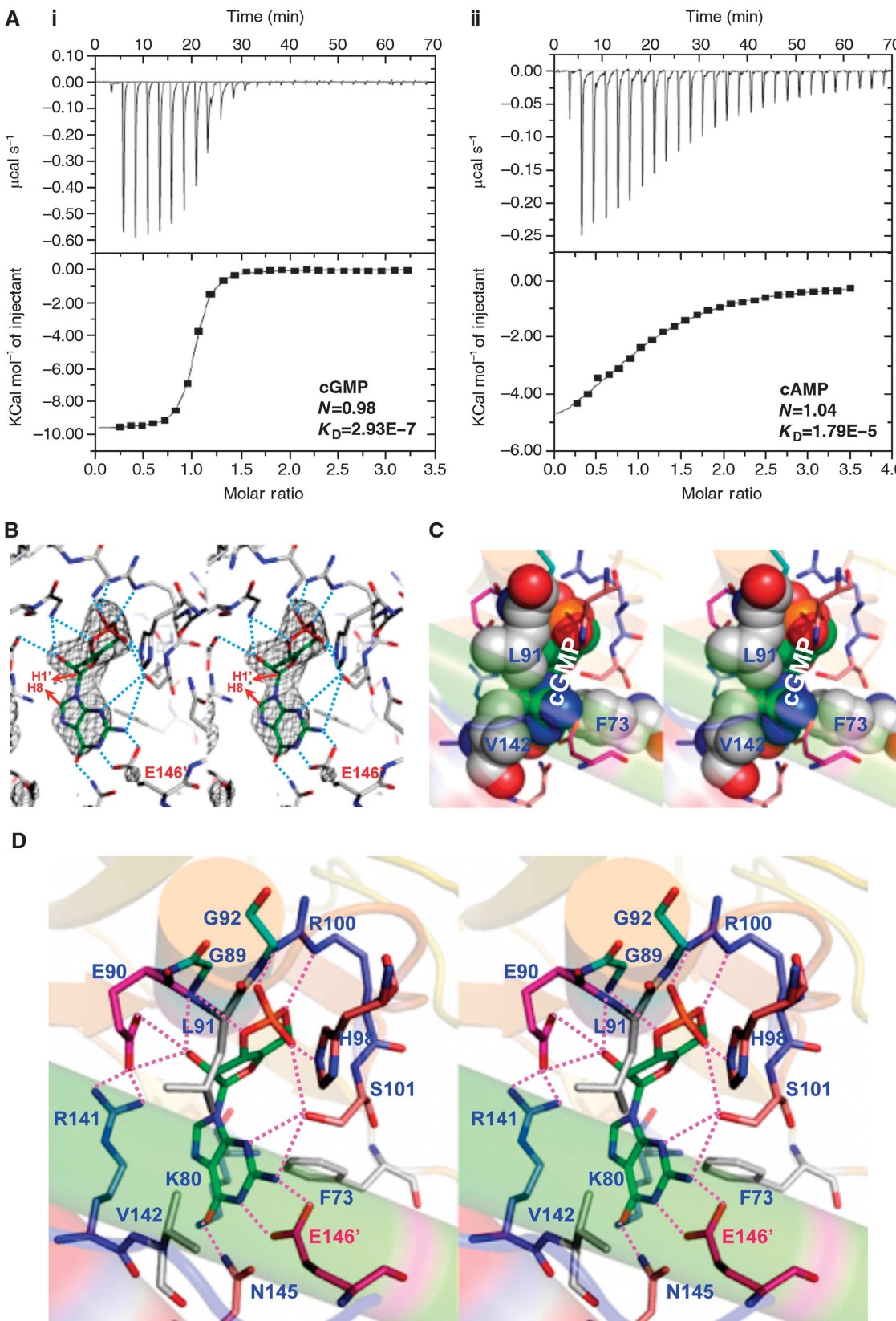

**Figure 4** Isothermal titration calorimetry and crystal structural analysis reveals XC_0249 to be a cyclic GMP-binding protein. (**A**) Isothermal titration calorimetry analysis of 20 μM XC_0249 with different nucleotides in PBS buffer at 21°C (i) Cyclic GMP (ii) cyclic AMP outcomes. (**B**) A stereo view of the Fo-Fc electron density map of the XC_0249 cNMP domain-cyclic GMP holo-complex. The guanine H8 and ribose H1' hydrogen atoms are drawn by red arrows to reveal their *syn* relationship. (**C**) The van der Waals plot of the XC_0249 cNMP binding pocket for cyclic GMP. Only hydrophobic residues are drawn in sphere, while the other residues are represented in stick form. (**D**) Protein–ligand interactions in the XC_0249 ligand-binding pocket. The carbons of cyclic GMP are coloured in green. Dotted red lines represent hydrogen bonds (complete list of ligand interactions is detailed in Supplementary Table S2).

also a critical residue, as its side chain amide nitrogen atom forms one H bond with the guanine O6 atom. In addition, potential hydrophobic interactions are also involved in cyclic GMP binding; F73 is involved in a perpendicular C–H bond/purine base interaction, and L91 and V142 are involved in interactions at the other face of the ligand with the ribose and guanine base (Figure 4). The key residues involved in interactions with cyclic GMP are highlighted in red in Figure 4. The complete set of interactions and their intermolecular distances are listed in Supplementary Table S3. The XC_0249 cyclic NMP domain dimer exhibits an interface area of 1344 Å$^2$, and each cyclic GMP exhibits an interface area of $\sim$300 Å$^2$, giving a $\Delta G$ of $-2.9$ kcal mol$^{-1}$ as calculated using the PISA program (Krissinel and Henrick, 2007).

This structural analysis implicated a number of residues (to include F73, E90, H98, N145 and E146) in cyclic GMP binding. The role of these residues was tested by examination of the effects of alanine substitution on nucleotide binding. Variant proteins with these alterations had considerably reduced cyclic GMP binding, as determined by ITC or by use of the radiolabelled nucleotide (Supplementary Figure S3; Supplementary Table S2).

### The regulatory action of XC_0249 overlaps with that of XC_0250

The above findings suggest that XC_0249 is a candidate cyclic GMP effector in *Xcc*. In order to examine this hypothesis, the phenotypic and transcriptional effects of mutation of *XC_0249* and *XC_0250* (encoding the guanylate cyclase) were compared. Mutation of *XC_0249* led to alteration in biofilm phenotype and reduced virulence to plants, effects

similar to those seen after mutation of *XC_0250* (Figure 5). Furthermore, comparative transcriptome profiling revealed an extensive overlap of regulatory influence of XC_0249 and XC_0250, with >50% of genes being commonly regulated (Figure 5; Supplementary Table S1). These observations support the contention that the effects of XC_0250 and cyclic GMP are exerted, at least in part, by interaction with XC_0249.

### XC_0249 binds cyclic GMP at the cyclic NMP-binding domain to modulate cyclic di-GMP signalling

The ability of XC_0249 to bind cyclic GMP suggests the possibility that the regulatory action of this effector is exerted through modulation of the DGC activity of the attached GGDEF domain. In initial experiments to address this possibility, the effect of addition of cyclic GMP on the synthesis of cyclic di-GMP by purified XC_0249 was assessed. The findings showed that cyclic GMP addition led to an increase in the activity of XC_0249 in cyclic di-GMP synthesis (Figure 6).

We hypothesised that the effect on modulation of the DGC activity of XC_0249 is associated with binding of cyclic GMP to the cNMP domain. In order to test this hypothesis, we assessed the effect of cyclic GMP addition on the DGC activity of variant proteins with alanine substitutions of residues F73, E90, H98, N145 and E146 that could no longer bind cyclic GMP (Supplementary Figures S3 and S4). Importantly, the DGC activity of these variant proteins was not altered by addition of cyclic GMP (Supplementary Figure S4). Furthermore, these variant proteins were less effective than the wild type in restoring biofilm formation to the *XC_0249* mutant (Supplementary Figure S4).

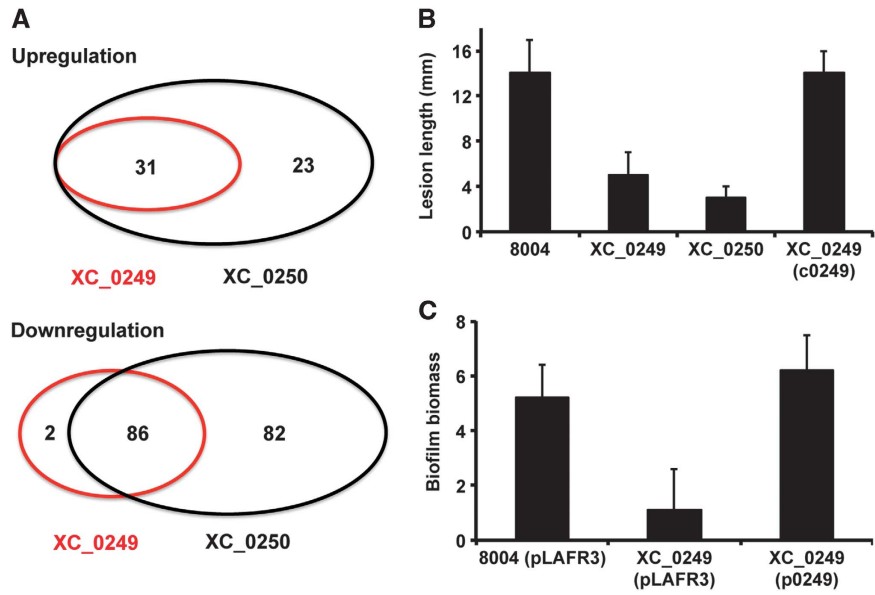

**Figure 5** Comparative transcriptome and phenotypic analyses reveal an extensive overlap in regulatory influence of XC_0249 and XC_0250. (**A**) Changes in gene expression in *XC_0249* and *XC_0250* deletion mutants compared with the wild-type *Xcc* as measured by RNA-Seq. Venn diagrams show the overlap of genes whose expression is either (i) upregulated or (ii) downregulated in the two mutant backgrounds. Values given are the mean and standard deviation of triplicate measurements (three biological and three technical replicates). Further detail of data analysis is given in Supplementary data. (**B**) Deletion of *XC_0249* affects virulence in Chinese Radish as shown by a decreased lesion length. The *Xcc* strains are 8004 (wild type), *XC_0249* deletion mutant and the *XC_0249* mutant carrying XC_0249 (c0249). The negative control was H$_2$O. Values are the means ± s.d. of 140 measurements for each strain. (**C**) The *XC_0249* mutant has decreased biofilm and cell adhesion compared to wild-type strain 8004 on a glass surface (biofilm biomass is a measure of absorbance at 550/600 nm). *In trans* expression of the gene encoding *XC_0249* (cloned as pXC_0249) in the *XC_0249* mutant restores these phenotypes towards wild type. Values given are the mean and standard deviation of triplicate measurements (three biological and three technical replicates).

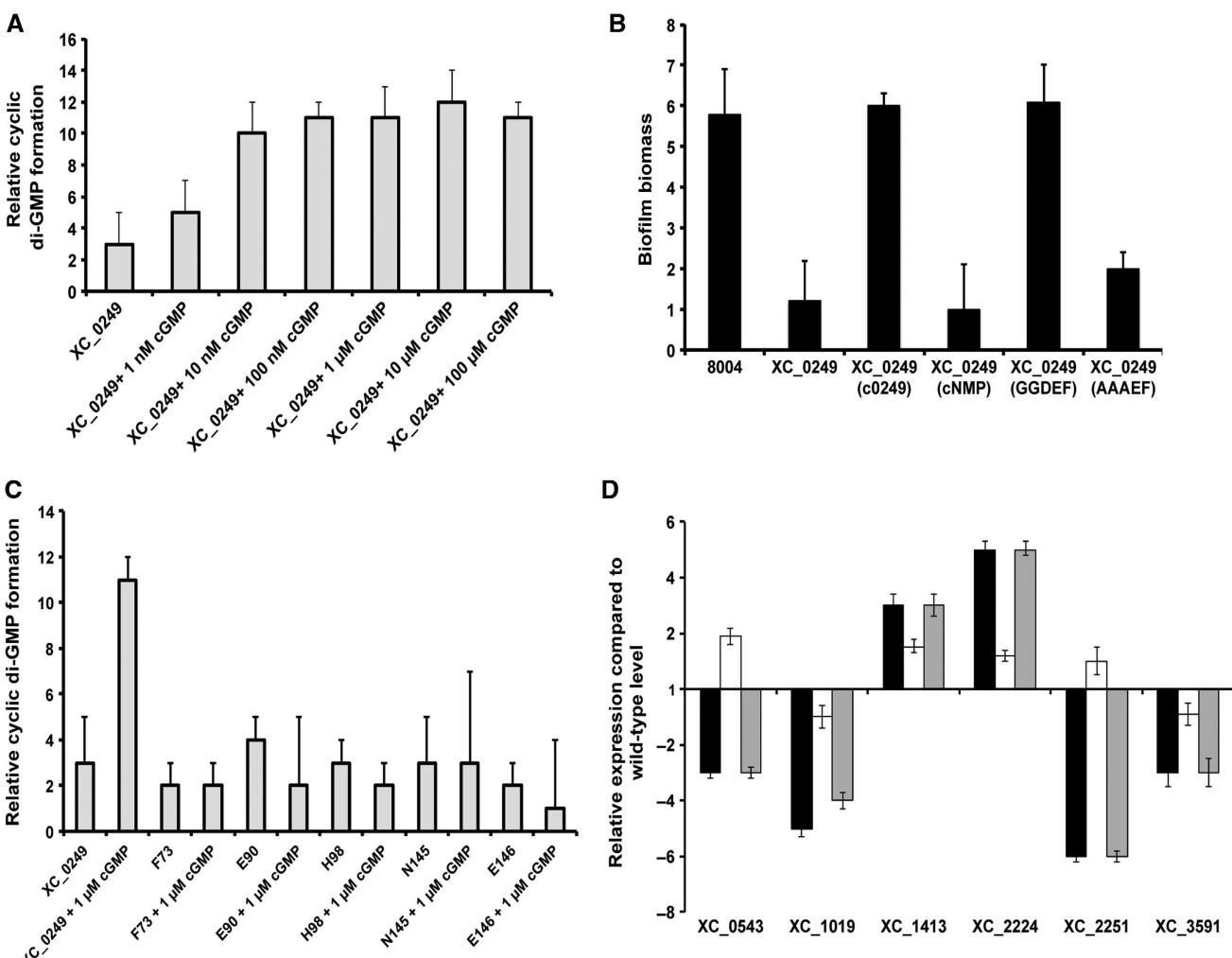

**Figure 6** The diguanylate cyclase activity of the XC_0249 protein is stimulated upon the binding of cyclic GMP and is required for regulation of gene expression. (**A**) Addition of increasing concentrations of cyclic GMP to the full-length purified XC_0249 protein led to an increase in the DGC activity of the enzyme. Values given are the mean and standard deviation of triplicate measurements (three biological and three technical replicates). (**B**) The enzymatic activity of variants of the full-length XC_0249 protein carrying alanine substitutions in residues involved in cyclic GMP binding (indicated in Supplementary Figure S3) was unaltered by the addition of cyclic GMP. Values given are the mean and standard deviation of triplicate measurements (three biological and three technical replicates). (**C**) Alanine substitution within the GGDEF domain (to give the AAAEF variant) abolishes the influence of XC_0249 on biofilm formation. Values given are the mean and standard deviation of triplicate measurements (three biological and three technical replicates). (**D**) Alanine substitution within the GGDEF domain (to give the AAAEF variant) abolishes the regulatory function of XC_0249 in gene expression. Quantitative RT–PCR analysis was used to measure transcript levels for *XC_0543* (TonB-dependent receptor), *XC_1019* (hypothetical protein), *XC_1413* (chemotaxis protein), *XC_2224* (secreted protein), *XC_2251* (sigma factor) and *XC_3591* (pectate lyase) in different *Xcc* strains. Data were normalised to 16S RNA and each is presented as the fold change with respect to the wild-type *Xcc* for each gene. Values given are the mean and standard deviation that are representative of four independent biological experiments. Black columns: effects of mutation of *XC_0249*; white columns: *XC_0249* mutant expressing a gene encoding *XC_0249* in a neutral site in the chromosome; grey columns: *XC_0249* mutant expressing a gene encoding *XC_0249* with a variant GGDEF domain.

## The regulation of biofilm formation and gene transcription by XC_0249 depends upon the DGC activity

The effect of cyclic GMP on the DGC activity of XC_0249 suggests that some regulatory influences of this protein may depend on the enzymatic activity. To test this contention, the effects of alanine substitution on several key residues required for enzymatic activity of the GGDEF domain (AAAEF) on the regulatory activity were assessed. For these experiments, the gene encoding the AAAEF variant of XC_0249 was incorporated into a neutral site in the chromosome of the *XC_0249* deletion mutant (see Materials and methods). Importantly, we have previously shown that this variant protein is expressed properly but lacks the DGC activity (Ryan *et al* 2010, 2012a).

Expression of the enzymatically inactive XC_0249 could not restore biofilm formation to the *XC_0249* deletion mutant (Figure 6). In addition, qRT–PCR showed that transcript levels for *XC_0543* (TonB-dependent receptor), *XC_1019* (hypothetical protein), *XC_1413* (chemotaxis protein), *XC_2224* (secreted protein), *XC_2251* (sigma factor) and *XC_3591* (pectate lyase) were unaltered by expression of the AAAEF variant in the *XC_0249* deletion mutant (Figure 6). These findings suggest that these regulatory activities of the protein depend upon its enzymatic activity.

## Discussion

The work in this paper provides evidence for a cyclic GMP-mediated pathway in *Xcc* that acts in the regulation of

virulence, biofilm formation and the transcription of specific genes. As far as we are aware, this is the first description of the role of cyclic GMP in bacterial biofilm formation or in the virulence of a bacterial pathogen. The synthesis of cyclic GMP in *Xcc* depends upon XC_0250, which we show is a guanylate cyclase with a class III nucleotidyl cyclase domain attached to a domain of uncharacterised function with tetra-tricopeptide repeats. Our data suggest that the action of cyclic GMP in *Xcc* depends in part on XC_0249, which has a cyclic GMP-binding domain attached to a GGDEF domain active in cyclic di-GMP synthesis. The action of XC_0249 thus represents a direct interplay between cyclic mononucleotide and cyclic dinucleotide signalling (Supplementary Figure S5).

Previous work has demonstrated links between cyclic mononucleotide and cyclic dinucleotide signalling in bacterial regulation, but these appear to be less direct than those involving XC_0249. Specifically, the cyclic AMP-dependent transcription factor CRP of *Vibrio cholerae* negatively controls the expression of *cdgA*, which encodes a DGC, in response to cyclic AMP (Fong and Yildiz, 2008). CdgA is required for biofilm formation by the *crp* mutant, suggesting that this regulatory interplay allows the bacterium to integrate information on nutrient availability status of the cell (reported by cyclic AMP) into decisions to undertake the biofilm lifestyle (Fong and Yildiz, 2008).

We have previously shown that the action of XC_0249 in the regulation of pilus-dependent motility in *Xcc* depends upon protein–protein interaction but not the enzymatic activity of the protein in cyclic di-GMP synthesis (Ryan *et al*, 2010, 2012a). Here, we show by contrast that the effects of XC_0249 on biofilm formation and transcription of specific genes does require the enzymatic activity. Overall, these findings indicate that some GGDEF domain proteins have a bi-functional nature. Comparative phenotypic and transcriptomic analysis of mutants shows that XC_0249 can only account for some of the regulatory effects of XC_0250 in *Xcc*. This knowledge indicates the existence of further effectors for cyclic GMP whose nature remains obscure. Equally, the cyclic di-GMP effectors that are involved in signal transduction beyond XC_0249 are not known. Some effectors of cyclic di-GMP signalling in *Xcc* have been described (Ryan *et al*, 2011), but it is likely that others remain to be discovered.

Recent studies have identified a cyclic GMP-dependent signalling system involving a transcription factor of the CRP-FNR protein family in the Alphaproteobacterium *R. centenum* that is required for encystment (Gomelsky, 2011; Marden *et al*, 2011). Although the pathway described here for *Xcc*, a Gammaproteobacterium, also serves to regulate a developmental process (that of biofilm formation), there are considerable differences between the elements of the two systems. The guanylate cyclases of *R. centenum* and *Xcc* show no significant amino-acid sequence similarity and the effectors of cyclic GMP action in *R. centenum* and *Xcc* are quite different, although both contain an N-terminal cyclic NMP-binding (cNMP) domain.

These observations raise the issue of what determines the selectivity of different cNMP domains for different nucleotides (Gomelsky, 2011). The cNMP domain of the *E. coli* CRP protein binds cyclic AMP preferentially over cyclic GMP. It is therefore interesting to compare the structure of the cyclic mononucleotide-binding domain of CRP with cyclic AMP bound with that of XC_0249 with cyclic GMP bound that

we describe here. The two structures can superimpose very well (with an r.m.s.d. of 1.37 Å out of 131 Cα atoms), except that the β3-β4 strands of XC_0249 are shorter than those of *Ec*CRP (Supplementary Figure S7). The major difference between the two structures is the diad motif of N145 and E146, which form strong H bonds with the guanine base of cyclic GMP (Supplementary Figure S7) and which contribute to cyclic GMP binding. Although this diad is conserved in all homologues of XC_0249 found in xanthomonads, it is not found in the cyclic NMP-binding domain in the *R. centenum* cyclic GMP effector. These findings may reflect the sequence and ligand flexibility noted among the CRP/FNR protein family (Gomelsky, 2011), as well as the substantial preference ($>60$-fold) of XC_0249 for binding cyclic GMP rather than cyclic AMP.

The work described here makes a contribution towards an understanding of cyclic GMP signalling that may extend beyond *Xcc*. Homologues of XC_0250 and XC_0249 that are also encoded by adjacent genes are found in a number of xanthomonads to include plant pathogens and the human pathogen *Stenotrophomonas maltophilia*. Furthermore, cyclase domains of related amino-acid sequence to XC_0250 are found in unrelated bacteria, including *Bradyrhizobium* and *Mycobacterium* spp. The involvement of these homologues in cyclic GMP signalling and their regulatory role in these other organisms remain to be tested.

## Materials and methods

Complete details of all methods used are provided in Supplementary Materials and methods.

### Bacterial strains, plasmids and culture conditions

The wild-type *Xcc* strain 8004 and mutants have been described previously (Slater *et al*, 2000; Ryan *et al*, 2007; An *et al*, 2013). For most experiments, *Xcc* strains were grown in NYGB medium, which comprises $5\,g\,l^{-1}$ bacteriological peptone (Oxoid), $3\,g\,l^{-1}$ yeast extract (Difco) and $20\,g\,l^{-1}$ glycerol. Other plasmids and strains used are described in Supplementary Table S4.

### Cyclic nucleotide cyclase assays

Adenylyl and guanylyl cyclase reactions were undertaken in $100\,\mu l$ reactions containing $0.5\,\mu g\,\mu l^{-1}$ purified protein of interest, $20\,mM$ Tris (pH 8.0), $100\,mM$ NaCl, $0.5\,mM$ ATP or GTP and $10\,mM$ MnCl$_2$ or MgCl$_2$. Reactions were incubated at $25^{\circ}$C, stopped by heating at $75^{\circ}$C for $10\,min$ and then clarified by centrifugation at $5000\,g$ for $10\,min$. High-pressure liquid chromatography to separate nucleotides was done as described previously (Ryan *et al*, 2010; Marden *et al*, 2011). The identity of nucleotides was confirmed by mass spectrometry. Cyclic di-GMP cyclase reactions were carried out using a standard enzymatic reaction mixture (total volume, $200\,\mu l$) that contained $5\,\mu M$ enzyme in $50\,mM$ Tris–HCl (pH 7.6), $10\,mM$ MgCl$_2$, $0.5\,mM$ EDTA and $50\,mM$ NaCl. The reaction was started by the addition of $50\,\mu l$ of GTP (final concentration, $150\,\mu M$) to the pre-warmed reaction mixture and was carried out for $60\,min$. The mixture was immediately placed in a boiling water bath for $3\,min$, followed by centrifugation at $15\,000\,g$ for $2\,min$. The supernatant was filtered through a $0.22$-μm-pore-size filter and analysed by high-pressure liquid chromatography (Ryan *et al*, 2006, 2010). The identity of cyclic di-GMP as a product was confirmed by mass spectrometry.

### RNA extraction and preparation

Three independent cultures of each selected *Xanthomonas* strain were subcultured and grown to logarithmic phase (0.7–0.8 OD$_{600}$) at $30^{\circ}$C in NYGB broth without selection. In all, $800\,\mu l$ of RNA protect (Qiagen) was added to $400\,\mu l$ culture and incubated at room temperature for $5\,min$. Cell suspensions were centrifuged, the supernatant was discarded and pellets were stored at $-80^{\circ}$C. After thawing, $100\,\mu l$ of TE-lysozyme ($400\,\mu g\,ml^{-1}$) was added

and the samples were incubated at room temperature. Total RNA was isolated using the RNeasy Mini Kit (Qiagen) whereby cells were homogenised utilising a 20-gauge needle and syringe. Samples were treated with DNase (Ambion) according to the manufacturer's instructions and the removal of DNA contamination was confirmed by PCR. A full description of RNA-Seq experiments and data analysis can be found in Supplementary Materials and methods.

### Virulence assays

The virulence of *Xcc* to Chinese Radish was estimated after bacteria were introduced into the leaves by leaf clipping as previously detailed (Dow *et al*, 2003; Ryan *et al*, 2007; An *et al*, 2013). Bacteria grown overnight in NYGB medium were washed and re-suspended in water to an OD at 600 nm of 0.001. For leaf clipping, the last completely expanded leaf was cut with scissors dipped in the bacterial suspensions. Thirty leaves were inoculated for each strain tested. Lesion length was measured 14 days after inoculation. Each strain was tested in at least four separate experiments.

### Statistical analyses

Statistical analyses were performed using the MINITAB software. Comparisons were done using one-way analysis of variance (ANOVA).

### Accession codes

Protein Data Bank: the coordinates and structure factors are deposited under the accession code (4KG1).

### Supplementary data

Supplementary data are available at *The EMBO Journal* Online (http://www.embojournal.org).

## Acknowledgements

The work of the authors has been supported in part by grants awarded by the Science Foundation of Ireland (SFI 07/IN.1/B955 to JMD and SFI 09/SIRG/B1654 to RPR), the Wellcome Trust (WT093314MA project grant to JMD and RPR and WT100204AIA senior fellowship grant to RPR) and National Science Council grants (97-2113-M005-005-MY3 to SHC). Work in SHC laboratory is supported in part by the Ministry of Education, Taiwan, ROC under the ATU plan, and by the National Science Council, Taiwan, ROC (grants 97-2113-M005-005-MY3). The National Synchrotron Radiation Research Center is a user facility supported by the National Science Council, Taiwan, ROC.

*Author contributions*: RPR, SC and JMD conceived and designed the experiments. RPR, SA, KC, MF, YM, JY, CL and DS performed the experiments. MF, DS and JR contributed new analytical tools. RPR, SA, KC, MF, YM, JY, CL, DS, JR, JMD and SC analysed the data. RPR, JMD and SC wrote the paper.

## Conflict of interest

The authors declare that they have no conflict of interest.

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
