## [Review Process File · The EMBO Journal]

Manuscript EMBO-2013-85624

A cyclic GMP-dependent signaling pathway regulates bacterial phytopathogenesis

Shi-Qi An, Ko-Hsin Chin, Melanie Febrer, Yvonne McCarthy, Jauo-Guey Yang, Chung-Liang Liu, David Swarbreck, Jane Rogers, J Maxwell Dow, Shan Ho Chou

Corresponding author: Robert Ryan, University of Dundee

Review timeline:

Submission date:	13 May 2013
Editorial Decision:	06 June 2013
Revision received:	18 June 2013
Editorial Decision:	01 July 2013
Accepted:	04 July 2013

Transaction Report:

Editor: David del Alamo

1st Editorial Decision

06 June 2013

Thank you for the submission of your manuscript entitled "A cyclic GMP-dependent signaling pathway regulates bacterial phytopathogenesis" to The EMBO Journal. We have now received the set of reports from the two referees asked to evaluate it, which I copy below.

As you can see from their comments, both referees are rather positive and recommend the publication of your manuscript. In general, they are convinced that the evidence presented properly supports your conclusions, and ask mainly for minor text and technical clarifications. Although these concerns are explicitly mentioned in the referee reports and thus I will not repeat them here, I would like to draw your attention in particular to points 1 and 2 of Referee #1 regarding certain missing controls in some of your experiments that may require additional experimental work.

Given these positive evaluations, I would like to invite you to submit a revised version of the manuscript. Please be aware that your revised manuscript must address the referees' concerns, experimentally if required, and their suggestions should be taken on board. It is 'The EMBO Journal' policy to allow a single round of revision only and, therefore, acceptance or rejection of your study will depend on the completeness of your responses included in the next, final version of the manuscript.

We generally allow three months as standard revision time. As a matter of policy, competing manuscripts published during this period will not be taken into consideration in our assessment of the novelty presented by your study ("scooping" protection). However, we request that you contact me as soon as possible upon publication of any related work in order to discuss how to proceed. Should you foresee a problem in meeting this three-month deadline, please let us know in advance and we may be able to grant an extension.

When preparing your letter of response to the referees' comments, bear in mind that this will form part of the Review Process File, and will therefore be available online to the community. For more details on our Transparent Editorial Process initiative, please visit our website: <http://www.nature.com/emboj/about/process.html>

Do not hesitate to contact me by e-mail or on the phone in case you have any questions or need further input.

Thank you for the opportunity to consider your work for publication. I look forward to your revision.

REFEREE REPORTS

Referee #1

"A cyclic GMP-dependent signaling pathway regulates bacterial phytopathogenesis"

Recently, research has shown that small nucleotide molecules play critical roles in bacterial signaling pathways. These pathways are of particular importance to both human and environmental health given their roles in regulating biofilm formation, pathogenicity and virulence. Previous research has established cyclic di-GMP and cyclic AMP as key players in the regulation of such pathways, however the role of cyclic GMP remains poorly understood. The manuscript by An et al. details a novel cyclic GMP signaling system found in the causative agent of black rot disease, *Xanthomonas campestris* (Xcc), which is involved in biofilm formation and plant pathogenicity. The authors identify two novel enzymes in Xcc involved in this signaling pathway- a guanylate cyclase and a di-guanylate cyclase whose activity is regulated by cyclic GMP. These observations provide for the first time a regulatory pathway that links cyclic GMP with biofilm formation and virulence, as well as cyclic GMP with cyclic di-GMP in the same signaling pathway. The authors point out that homologs of these two enzymes are found in a variety of other bacteria, suggesting this novel cyclic GMP-mediated pathway may be a more general signaling mechanism. Overall, the manuscript is well written and the general conclusions are supported by the data. Nevertheless, there are a few concerns that deserve attention. Addressing these would strengthen the manuscript significantly.

1. The authors assert that complementation of XC_0250-deficient Xcc with the enzymatically inactive GGDEF domain of XC_0250 (the AAAEF mutant) does not rescue biofilm formation. Do the authors know that this multiple mutation variant folds properly? This is important for the conclusion that the regulatory activities depend on enzymatic activity. Additionally, including an activity measurement of both the GGDEF and AAAEF proteins in Fig. 6B would provide convincing controls to strengthen this argument.
2. The lack of expression controls of wild-type and mutant proteins in the various genetic backgrounds is a general concern throughout the manuscript. These controls may also reveal whether the ectopically expressed proteins and the mutant variants are expressed at the same level, and similar to native protein expression in the unmodified strain. This is common practice and especially important for studies with loss-of-function as the predominant phenotype.
3. Based on Figs. 2C and 2D, and the values reported in the Materials and Methods, the average in vitro activity of the XC_0250 is quite low (~1 turnover per active site every 10 minutes by the reviewer's calculation assuming a linear reaction rate). The authors might want to address this observation in terms of its physiologic relevance and in relation to activities reported for other guanylyl cyclases. Such a low activity could be explained by a non-linear reaction rate caused by product inhibition, which could easily be tested by incubating XC_0250 in the presence of cyclic GMP before the addition of GTP. Observing product inhibition could be important to this manuscript as it would provide an additional level of regulatory control in this pathway. Along the same lines, the fact that XC_0249 has "full" activity when using 10 nM cyclic GMP (even though the K_d for cyclic GMP is 30 times that concentration) might also suggest some form of product inhibition with this enzyme as well. Finally, it would be a valuable addition to the manuscript if the authors could demonstrate changes in cellular cyclic di-GMP levels in the XC_0249 and XC_0250 deletion strains (compared to the wild-type strain).

4. Page 5, line 6: Reactions with ATP are not shown. Either include the data in the Figure or mention that this is "data not shown".
5. How were the gene products shown in Fig. 3C chosen, given the dozens reported in Table S1? Without an explanation, it would seem that they were chosen simply because the RNA-seq results matched quite nicely with those from qRT-PCR, in which case the overall utility of this figure (and any conclusions regarding global correlations between RNA-seq and qRT-PCR results) must be brought into question. Please provide a rationale.
6. Page 7 lines 13-17: Please discuss and illustrate the dimeric interactions of XC_0249 cyclic-GMP binding domain. Is this protein a dimer in solution? Without further discussion about the dimeric nature of this protein and its importance in function, such analysis would seem superfluous. Also, a short comparison in the Results section with the closest known structurally characterized homologs using the DALI server would give readers a better sense of the novelty of this structure.
7. Kd values reported on page 6, lines 21-22 should have errors associated with them.
8. Figure 2A legend should include the contents of each lane.
9. Figure Legend 2: Please report the molar concentration of enzyme used in the assays.
10. Figure 4B legend should include the contour level of the displayed Fo-Fc map. Also indicate in the legend that these are stereoview images.
11. Supplementary Fig. 4 legend has duplicate sentences. Also, "E09A" should be "E90A".
12. Graphs displaying biofilm formation results have no units associated with the y-axis.
13. Page 11, line 1: "Supplementary Table S3" should be "Figure S6A".
14. Page 11, Discussion: Please provide rmsd values for the superposition. Also, please include a figure illustrating the superposition using the crystal structures.
15. Please discuss a model of how cyclic GMP binding to XC_0249 may allosterically regulate diguanylate cyclase activity.

Referee #2

This is a very exciting paper. My comments deal primarily with the presentation issues.

1. In my opinion, a key accomplishment of this work is determination of the 3D structure of the cGMP-specific cNMP_BINDING (Pfam PF00027, SMART SM00100) domain in complex with the ligand. Surprisingly, this result is not even mentioned in the Abstract. The Discussion section says (pp. 10-11) "It is therefore interesting to compare the structure of the cyclic mononucleotide-binding domain with cyclic AMP bound with that of XC_0249 bound to cyclic GMP that we describe here. The two structures can superimpose very well (Supplementary Table S3)...". First, there is no superposition data in Supplementary Table S3, it only shows statistics of data collection and structural refinement. Second, a comparison of cAMP- and cGMP-binding structures would indeed be extremely interesting. Why hasn't this been done? A figure showing a superposition of the two structures would definitely increase the appeal of the whole paper.
2. p.10, l. 24. "the effectors of cyclic GMP in R. centenum and Xcc are quite different". As far as I understand, the effectors of cGMP in either case are unknown. You probably meant "the outputs of cGMP signaling" which are different indeed.
3. In Fig. 1, the transposon insertion mutant is marked as MT0479 in panel A and as MT0478 in panel C. The domain architecture representation taken from the SMART database (Letunic et al., 2012) should be annotated as such and the appropriate reference must be provided.
4. Two columns out of four in Fig. 5B are the same as in Fig. 3Aii. Why not combine these into a single plot?
5. It would be helpful to combine all supplementary materials into a single PDF file or two. The reader should not be expected to download 11 separate supplementary files.

6. In Fig. S1A, it would be helpful to label predicted active-site residues of XC0250 (at least those mutated in panel B).
7. The legend to Fig. S2A should mention that its domain architecture is taken from the SMART database (Letunic et al., 2012).
8. Fig. S5 shows XC_0250 as a membrane-anchored protein. There are no predicted membrane-spanning domains in the XC_0250 sequence and both CYC and TPR domains have been found so far exclusively in the cytosol. After the figure is corrected, it might deserve to be moved into the main text.

1st Revision - authors' response

18 June 2013

A point-for-point rebuttal to referees comments

Referee #1:

1. The authors assert that complementation of XC_0250-deficient Xcc with the enzymatically inactive GGDEF domain of XC_0250 (the AAAEF mutant) does not rescue biofilm formation. Do the authors know that this multiple mutation variant folds properly? This is important for the conclusion that the regulatory activities depend on enzymatic activity. Additionally, including an activity measurement of both the GGDEF and AAAEF proteins in Fig. 6B would provide convincing controls to strengthen this argument.

Author response: We have confirmed in two previous studies [please see Ryan et al., (2010) Proc Natl Acad Sci U S A. 107(13): 5989-94. and Ryan et al., (2012) Mol Microbiol. 86(3): 557-67] that the enzymatically inactive GGDEF domain of XC_0249 is expressed at the same level as the wild type, through the use of Western blotting. Furthermore, the enzymatically-inactive version still retains the ability to control motility. These findings are consistent with the contention the AAAEF variant folds properly. In these reports we also provided documentation of the enzymatic activity for both the GGDEF and AAAEF variant proteins. For this reason we do not believe it necessary to include this data again in this paper. However, we do agree with the referee that this point needs to be stated in the current manuscript and have include in the Results (Page 9 line 5) the sentence, "Importantly, we have previously shown that this variant protein is expressed properly but lacks DGC activity (Ryan *et al* 2010; 2012a)" to address this.

2. The lack of expression controls of wild-type and mutant proteins in the various genetic backgrounds is a general concern throughout the manuscript. These controls may also reveal whether the ectopically expressed proteins and the mutant variants are expressed at the same level, and similar to native protein expression in the unmodified strain. This is common practice and especially important for studies with loss-of-function as the predominant phenotype.

Author response: The referee is correct to assert that it is common practice to include expression controls and we have indeed done this, both here and in previous papers. We have now included statements throughout the article to indicate this to the reader. As an example please see Page 5 line 17, "Importantly, western analysis showed that all variant proteins were expressed in XC_0250 deletion mutant and wild-type backgrounds".

3. Based on Figs. 2C and 2D, and the values reported in the Materials and Methods, the average in vitro activity of the XC_0250 is quite low (~1 turnover per active site every 10 minutes by the reviewer's calculation assuming a linear reaction rate). The authors might want to address this observation in terms of its physiologic relevance and in relation to activities reported for other guanylyl cyclases. Such a low activity could be explained by a non-linear reaction rate caused by product inhibition, which could easily be tested by incubating XC_0250 in the presence of cyclic GMP before the addition of GTP. Observing product inhibition could be important to this manuscript as it would provide an additional level of regulatory control in this pathway. Along the same lines, the fact that XC_0249 has "full" activity when using 10 nM cyclic GMP (even though the Kd for cyclic GMP is 30 times that concentration) might also suggest some form of product inhibition with this enzyme as well. Finally, it would be a valuable addition to the manuscript if the authors could demonstrate changes in cellular cyclic di-GMP levels in the XC_0249 and XC_0250 deletion strains (compared to the wild-type strain).

Author response: We agree that the in vitro cyclase activity of XC_0250 is quite low compared to other studied eukaryotic cGMP cyclases under the conditions that we tested. However, it is important to note that the in vitro conditions we used are described as optimal for eukaryotic guanylyl cyclases [see Danchin, A. (1993) *Adv Second Messenger Phosphoprotein Res* 27: 109–162] but may not be for XC_0250. In addition the influence on the overall activity of the protein of the uncharacterized domain with tetratricopeptide repeats that is attached to the cyclase domain is unclear. Consequently we would not wish to comment on the physiologic relevance of the level of activity and relation to activities reported for other guanylyl cyclases, as we consider it would be premature to do so.

As the referee suggested we have measured (but not reported) the “global” cellular level of cyclic di-GMP in both XC_0249 and XC_0250 deletion strains grown under biofilm and planktonic conditions. We saw no significant difference in cellular levels of cyclic di-GMP compared to the wild-type strain. It is not however uncommon in the field of cyclic di-GMP signalling to see an alteration in phenotype after loss by mutation of a signalling protein without observing an accompanying measurable alteration in the level of the nucleotide. Some researchers have suggested that this means that the signalling protein influences localized ‘pools’ of the nucleotide. Although our observations point to a regulatory mechanism of XC_0249 that is more subtle than modulation of the “global” levels of cyclic di-GMP in Xcc, we feel that this is not an issue we would wish to address in the current study.

4. Page 5, line 6: Reactions with ATP are not shown. Either include the data in the Figure or mention that this is "data not shown".

Author response: We have not reported the ATP data in the Figure and therefore included the statement “data not shown” in the text. Please see, “This protein had guanylate cyclase activity, converting GTP to cyclic GMP but had no activity as an adenylate cyclase on ATP (Fig 2 and data not shown)”.

5. How were the gene products shown in Fig. 3C chosen, given the dozens reported in Table S1? Without an explanation, it would seem that they were chosen simply because the RNA-seq results matched quite nicely with those from qRT-PCR, in which case the overall utility of this figure (and any conclusions regarding global correlations between RNA-seq and qRT-PCR results) must be brought into question. Please provide a rationale.

Author response: The rationale for selection of the genes in Fig. 3C was that the gene (or gene product) had been shown previously to have a role in virulence in Xcc. We have included a statement to this effect in the Results section to clarify this point. Please see (Page 6 line 8), “The effect of XC_0250 mutation on the level of transcript was validated by quantitative reverse transcription polymerase chain reaction (qRT-PCR) analysis for a panel of genes whose members were selected because of a previous implication in the virulence of Xcc (Figure 3C).

6. Page 7 lines 13-17: Please discuss and illustrate the dimeric interactions of XC_0249 cyclic-GMP binding domain. Is this protein a dimer in solution? Without further discussion about the dimeric nature of this protein and its importance in function, such analysis would seem superfluous. Also, a short comparison in the Results section with the closest known structurally characterized homologs using the DALI server would give readers a better sense of the novelty of this structure.

Author response: Like most bacterial diguanylate cyclases, XC_0249 appears to form a dimer *in vitro* and it is established that DGCs are active in cyclic di-GMP synthesis as dimers. Therefore, we feel no need to edit this section of text. We describe a comparison of the structure determined here with the cNMP-binding domain of CRP of *E. coli* in the Discussion section pages 10-11 and in supplementary Figure S7. We feel that this addresses the issue of the novelty, since the two structures can superimpose very well.

7. Kd values reported on page 6, lines 21-22 should have errors associated with them.

Author response: As requested we have included the standard errors associated with Kd values. Please see (Page 6 line 21), “The findings showed that the cNMP domain of XC_0249 bound cyclic GMP with a high affinity (K_D of $0.293 \pm 0.16 \mu\text{M}$), but had lower affinity for cyclic AMP ($17.9 \pm 2.02 \mu\text{M}$), cyclic di-GMP or cyclic di-AMP (Figure 4A; Supplementary Table S2)”.

8. Figure 2A legend should include the contents of each lane.

Author response: We have amended legend to Fig 2A to include the appropriate information.

9. Figure Legend 2: Please report the molar concentration of enzyme used in the assays.

Author response: We have amended legend to Fig 2 to include this information.

10. Figure 4B legend should include the contour level of the displayed Fo-Fc map. Also indicate in the legend that these are stereoview images.

Author response: We have amended Fig. 4 to include the contour level of the displayed Fo-Fc and to highlight that the images are in fact a stereo view representation.

11. Supplementary Fig. 4 legend has duplicate sentences. Also, "E09A" should be "E90A".

Author response: This is a typo and we have amended Fig. 4 legend to correct this. The legend now reads, "A variant in the cNMP of XC_0249 that no longer bound cyclic GMP (F73A, E90A) was unresponsive to the addition of cyclic GMP and less effective than the wild-type in restoring biofilm formation to the XC_0249 mutant (B)"

12. Graphs displaying biofilm formation results have no units associated with the y-axis.

Author response: We have included biofilm biomass units in the legends to Figure 3 and Figure 5 to rectify this.

13. Page 11, line 1: "Supplementary Table S3" should be "Figure S6A".

Author response: This is an error which we have amended.

14. Page 11, Discussion: Please provide rmsd values for the superposition.

Author response: We provide rmsd values for the superposition. The text now reads, "It is therefore interesting to compare the structure of the cyclic mononucleotide-binding domain with cyclic AMP bound with that of XC_0249 bound to cyclic GMP that we describe here. The two structures can superimpose very well (with a rmsd of 1.37 Å out of 131 C α atoms), except that the β 3- β 4 strands of XC_0249 are shorter than those of EcCRP (Figure S7)."

15. Please discuss a model of how cyclic GMP binding to XC_0249 may allosterically regulate diguanylate cyclase activity.

Author response: We speculate that the binding of cyclic GMP may promote conformational changes allowing the juxtaposition of two GGDEF domains in a configuration productive for cyclic di-GMP synthesis. However in the absence of structural information on the full-length protein, we would not wish to include such speculation on the mechanism of DGC activation in this manuscript.

Referee #2

This is a very exciting paper. My comments deal primarily with the presentation issues.

1. In my opinion, a key accomplishment of this work is determination of the 3D structure of the cGMP-specific cNMP_BINDING (Pfam PF00027, SMART SM00100) domain in complex with the ligand. Surprisingly, this result is not even mentioned in the Abstract.

The Discussion section says (pp. 10-11) "It is therefore interesting to compare the structure of the cyclic mononucleotide-binding domain with cyclic AMP bound with that of XC_0249 bound to cyclic GMP that we describe here. The two structures can superimpose very well (Supplementary Table S3)...". First, there is no superposition data in Supplementary Table S3, it only shows statistics of data collection and structural refinement. Second, a comparison of cAMP- and cGMP-binding structures would indeed be extremely interesting. Why hasn't this been done? A figure showing a superposition of the two structures would definitely increase the appeal of the whole paper.

Author response: We have edited the abstract to mention the determination of the 3D structure of the cGMP-specific cNMP-binding domain. Please see, "The isolated cNMP domain of XC_0249 bound cyclic GMP and a structure-function analysis, directed by determination of the crystal structure of the holo-complex, demonstrated the site of cyclic GMP binding that modulates cyclic di-GMP synthesis"

In the revision of the manuscript we have included a new supplementary figure describing the superposition of the structure of the cyclic mononucleotide-binding domain of XC_0249 with cyclic GMP bound and the cyclic mononucleotide-binding domain of *E. coli* CRP with cyclic AMP bound. The modified text now reads, "It is therefore interesting to compare the structure of the cyclic mononucleotide-binding domain of CRP with cyclic AMP bound with that of XC_0249 with cyclic GMP bound that we describe here. The two structures can superimpose very well (with a rmsd of 1.37 Å out of 131 C α atoms), except that the β 3- β 4 strands of XC_0249 are shorter than those of EcCRP (Figure S7)."

Figure S7. A model showing the superimposition of the cNMP domain of XC_0249 (colored in red) and that of *E. coli* CRP (1G6N) (colored in marine). The central α C helices superimpose/fit strongly while the cNMP binding β -barrel domains deviate due to the binding of different cyclic-mononucleotide. (A). An expanded view of these regions show that the cyclic GMP (the carbons colored in magenta) adopts a *syn*-conformation (the G-H8 and ribose-H1' atoms are located in the same side), which is different to the *anti*-conformation exhibited by cyclic AMP (carbons colored in cyan) in the *E. coli* CRP complex structure (B). It is also important to note that strong H-binding occurs between the cyclic GMP G-O6 and G-2NH2 with the side chain atoms of Asn144 in the α C helix and Glu145' in the α C' helix, respectively. In contrast, in the *E. coli* CRP-cAMP complex, the cyclic AMP A-6NH2 binds with the Ser428 in the α C helix and S428' in the α C' helix.

2. p.10, l. 24. "the effectors of cyclic GMP in *R. centenum* and *Xcc* are quite different". As far as I understand, the effectors of cGMP in either case are unknown. You probably meant "the outputs of cGMP signaling" which are different indeed.

Author response: We use the term 'effector of cyclic GMP' to mean those proteins that are receptors that can bind the nucleotide second messenger to elicit or affect a particular outcome. We think that the reviewer has taken this term to mean those molecules or signals that lead to an alteration in cyclic GMP. To avoid any ambiguity here we have reworded the sentence to, "The guanylate cyclases of *R. centenum* and *Xcc* show no significant amino acid sequence similarity and the effectors of cyclic GMP action in *R. centenum* and *Xcc* are quite different, although both contain an N-terminal cyclic NMP-binding (cNMP) domain".

3. In Fig. 1, the transposon insertion mutant is marked as MT0479 in panel A and as MT0478 in panel C. The domain architecture representation taken from the SMART database (Letunic et al., 2012) should be annotated as such and the appropriate reference must be provided.

Author response: This is a typo that has been amended. We have also included a citation to the SMART database as instructed.

4. Two columns out of four in Fig. 5B are the same as in Fig. 3Aii. Why not combine these into a single plot?

Author response: We feel that having all the data presented in Fig 5B would make the figure to dense for readers. We prefer to leave the data presented over two figures.

5. It would be helpful to combine all supplementary materials into a single PDF file or two. The reader should not be expected to download 11 separate supplementary files.

Author response: We agree with the referee here and in the resubmission have generated a single PDF file of all supplementary material.

6. In Fig. S1A, it would be helpful to label predicted active-site residues of XC0250 (at least those mutated in panel B).

Author response: We have highlighted the predicted active-site residues of XC_0250 in revised Fig S1A.

7. The legend to Fig. S2A should mention that its domain architecture is taken from the SMART database (Letunic et al., 2012).

Author response: We have included a citation to the SMART database in the revised legend to Fig S2.

8. Fig. S5 shows XC_0250 as a membrane-anchored protein. There are no predicted membrane-spanning domains in the XC_0250 sequence and both CYC and TPR domains have been found so

far exclusively in the cytosol. After the figure is corrected, it might deserve to be moved into the main text.

Author response: Some programs such as TOP PRED and TM Pred do suggest the occurrence of membrane-spanning sequences within XC_0250. However in the absence of experimental data to determine topology, we have amended the legend to Figure S5 to include the statement “Although XC_0250 is depicted as a membrane-anchored protein, it is also possible that both CYC and TPR-containing domains are in the cytosol”. We would prefer to keep this Figure in the Supplementary Information.

2nd Editorial Decision

01 July 2013

Thank you for the submission of the revised version of your manuscript entitled "A cyclic GMP-dependent signaling pathway regulates bacterial phytopathogenesis" and please accept my apologies for the delay in responding. As you have properly dealt with the concerns originally raised by the referees, I am writing with an 'accept in principle' decision, which means that I will be happy to accept your manuscript for publication once a few more details have been addressed, as follows.

Browsing through the manuscript, I have noticed minor issues with the description of your statistical analyses, particularly with the definition of the error bars used in some panels and the number of experiments that they represent. This specifically applies to parts of figures 3, 5, 6, S1, S2 and S4. As a guide, statistical analyses must be described either in the Materials and Methods section or in the legend of the figure to which they apply and will include a definition of the error bars used and the number of independent experiments performed.

Additionally, the western blot presented in figure 2A shows signs of splicing. Although splicing of gels and blots is allowed in order to avoid unnecessary waste of space, it should be clearly stated in the legend and marked in the figure with lines separating the originally non-contiguous lanes. In this regard, I would also like to mention that, as a novel initiative in The EMBO Journal, we now encourage the publication of source data, particularly for electrophoretic gels and blots, with the aim of making primary data more accessible and transparent to the reader. Although optional at the moment, I would like to suggest that you provide the full blot as a PDF file for further clarity. The PDF file should be labeled "figure 2A", and should have molecular weight markers; further annotation could be useful but is not essential. The file would be published online with the article as a supplementary "Source Data" file. If you have any questions regarding this initiative do not hesitate to contact me.

Thank you very much for your patience and congratulations in advance on a successful publication. Once these minor changes are incorporated into the manuscript, you will receive an official acceptance letter with further instructions on how to proceed with the publication process.